# Physical and Chemical Characterisation of Conventional and Nano/Emulsions: Influence of Vegetable Oils from Different Origins

**DOI:** 10.3390/foods11050681

**Published:** 2022-02-25

**Authors:** Jansuda Kampa, Richard Frazier, Julia Rodriguez-Garcia

**Affiliations:** Department of Food and Nutritional Sciences, University of Reading, Whiteknights, Reading RG6 6DX, UK; j.kampa@pgr.reading.ac.uk (J.K.); r.a.frazier@reading.ac.uk (R.F.)

**Keywords:** refined vegetable oil, virgin vegetable oil, fatty acids, antioxidants, creaming index, lipid oxidation, emulsion, nanoemulsion

## Abstract

The processes of oil production play an important role in defining the final physical and chemical properties of vegetable oils, which have an influence on the formation and characteristics of emulsions. The objective of this work was to investigate the correlations between oils’ physical and chemical properties with the stability of conventional emulsions (d > 200 nm) and nanoemulsions (d < 200 nm). Five vegetable oils obtained from different production processes and with high proportion of unsaturated fatty acids were studied. Extra virgin olive oil (EVOO), cold-pressed rapeseed oil (CPRO), refined olive oil (OO), refined rapeseed oil (RO) and refined sunflower oil (SO) were used in this study. The results showed that the physicochemical stability of emulsion was affected by fatty acid composition, the presence of antioxidants, free fatty acids and droplet size. There was a significant positive correlation (*p* < 0.05) between the fraction of unsaturated fatty acids and emulsion oxidative stability, where SO, OO and EVOO showed a significantly higher lipid oxidative stability compared to RO and CPRO emulsions. Nanoemulsions with a smaller droplet size showed better physical stability than conventional emulsions. However, there was not a significant correlation between the oxidative stability of emulsions, droplet size and antioxidant capacity of oils.

## 1. Introduction

In recent decades, there has been an increasing interest from consumers, food industry and health organisations in vegetable oils with high content of unsaturated fatty acids due to their effect on human health compared to saturated fatty acids. The consumption of saturated fatty acids increases the risk of cardiovascular diseases and increases low-density lipoprotein (LDL) and cholesterol in blood [1,2]. World Health Organization and other governmental organisations have consistently highlighted the importance of controlling saturated fat and trans-fat content in food products in order to decrease disease burden in the population [3,4]. However, the replacement of saturated fat by vegetable oils with high content of unsaturated fatty acids is very challenging due to the specific technological functionality of saturated fat in food products. The fatty acid composition influences their lipid oxidation rate [5,6]; oils with high content of unsaturated fatty acids are more susceptible to lipid oxidation, which can lead to undesirable flavour profile, texture, shelf life and loss of nutritional quality in food products [7,8]. In fact, certain operations used as part of oil production process, such as roasting the seeds before pressing, improve the oxidative stability of oils [9,10].

There are several vegetable oils containing high level of unsaturated fatty acids. Monounsaturated fatty acids are mainly found in olive oil and rapeseed oil [11,12]; the major sources of polyunsaturated fatty acids are found in sunflower oil and flaxseed oil [12,13,14]. However, not only fatty acid composition of oil, but also oil production process has an influence on physical and chemical properties of oils. The difference in extraction and refining process of oil production can lead to a difference in the content of free fatty acids, colour compounds, total phenolic compounds and antioxidant capacity between refined and unrefined oils. Oil is generally extracted by pressing or by solvent extraction followed by refining of crude fats [15,16]. The refining process involves either chemical or physical refining, which includes several stages: degumming, neutralisation, bleaching and deodorisation [5]. The content of inherent antioxidants, including total phenolic and tocopherols, is a major influence on free radical scavenging activity [17,18] and oxidative stability in oils [19,20]. Pigments, natural antioxidants and free fatty acids are removed during degumming, bleaching and deodorisation stages of oil refining processes [11,21,22]. Therefore, the compositional differences in the oils, such as higher phenolic and free fatty content in unrefined oils or lower level of antioxidants and phenolic content in refined oils, could have an effect on the final properties and stability of emulsions elaborated with these oils to be used as saturated fat replacers.

Research has been carried out to develop strategies to replace saturated fat in food products using vegetable oils with high content of unsaturated fatty acids [23,24,25,26]. Several studies have focused on the effect of fatty acid chain length (long chain triglycerides, medium chain triglycerides and short chain triglycerides) on emulsion stability and bioaccessibility of lipophilic bioactives [27,28,29]; other work has been carried out on the effect of fatty acid profile of vegetable oils on characterisation and stability of emulsions [30,31,32]. Research has shown that the size of oil droplets has an effect on the physical and chemical stability of emulsions. Droplet diameter was influenced by the chain length of triglycerides: smaller droplets were obtained when using medium chain triglyceride rather than long and short chain triglycerides [28]. Nanoemulsions prepared from cold-pressed pomegranate seed oil had bigger droplets and lower emulsion stability because of its higher viscosity and proportion of unsaturated fatty acids in comparison to other oils [31]. Nanoemulsions, with oil droplet size of less than 200 nm [33], could provide better physical stability to gravitational separation of the fat phase [34] than conventional emulsions (oil particle size more than 200 nm) [35,36], improving physicochemical stability of functional compounds and unsaturated oils [33,37]. However, the smaller droplet size could promote lipid oxidation by accelerating reactions at the surface of the droplets; a high rate of lipid oxidation is attributed to a high droplet surface area, as the droplet size decreases [7,38].

Although there has been a great number of studies dealing with the effect of fatty acids profile of oils on emulsion properties, such as oil droplet size and lipid oxidation, there are no works that have studied the effect of oil production processes on the stability of conventional and nanoemulsions. The evaluation of correlations between oil composition due to origin and emulsion structure and stability will help elucidating the importance of oil selection for the production of emulsions with specific applications in the food industry. It is hypothesised that emulsions made with unrefined oils and with smaller droplet diameter will present better chemical and physical stability due to higher concentration of antioxidant components and a lower creaming mechanism. The objective of this study was to assess the physicochemical properties (mean droplet diameter and polydispersity index) and stability (creaming index and lipid oxidation) of emulsions made of five vegetable oils from different botanical origins (olive, rapeseed, sunflower) and different production processes (refined and unrefined). The effects of droplet size and distribution on conventional and nanoemulsion stability were also investigated to elucidate which oil should be selected for this type of colloidal structure.

## 2. Materials and Methods

### 2.1. Reagents and Standards

Tween 20 (Polyoxyethylene sorbitan monolaurate), with hydrophilic-lipophilic balance (HLB) value of 16.7, was used as food-grade nonionic surfactant. Fatty acid methyl esters (the standard reference material FAMEs included C8:0, C10:0, C12:0, C14:0, C16:0, C18:0, C18:1, C18:2, C18:3), boron trifluoride reagent (BF_3_) (13–15% *w*/*w* in methanol), methanol (≥99.9% *v*/*v*) (CH_3_OH), heptane (99% *v*/*v*) (CH_3_(CH_2_)_5_CH_3_), sodium chloride (NaCl), Folin–Ciocalteu reagent, gallic acid (C_6_H_2_(OH)_3_COOH), sodium carbonate (Na_2_CO_3_), diethyl ether, ethanol (95% *v*/*v*) ((CH_3_CH_2_)_2_O), potassium hydroxide (KOH), phenolphthalein (C_20_H_14_O_4_), 1,1-diphenyl-2-picrylhydrazyl radical (DPPH), trichloroacetic acid (≥99% *w*/*w*) (CCl_3_COOH) and thiobarbituric acid (98% *w*/*w*) (C_4_H_4_N_2_O_2_S) were purchased from Sigma-Aldrich Co., Ltd. (Dorset, UK). Sodium hydroxide (NaHO), anhydrous sodium sulphate (≥99% *w*/*w*) (Na_2_SO_4_) and hydrochloric acid (37% *w*/*w*) (HCI) were purchased from Fisher scientific Co., Ltd. (Loughborough, UK). High-purity water was used for the preparation and dilution of reagents and standards.

### 2.2. Oil Samples

Five oils were selected for the study due to their high content of unsaturated fatty acids, their different unsaturation profile and extraction processes: extra virgin olive oil (EVOO) (14.26% of saturated fat, 77.69% of monosaturated fat and 8.04% of polyunsaturated fat; Napolina brand, UK retail market), cold-pressed rapeseed oil (CPRO) (4.88% of saturated fat, 73.39% of monosaturated fat and 20.73% of polyunsaturated fat; Farrington’s mellow yellow brand, UK retail market), Olive oil (OO) (composed of refined olive oil and virgin olive oil, 13.28% of saturated fat, 79.73% of monosaturated fat and 6.98% of polyunsaturated fat; Brakes Bros Co., Ltd., Kent, UK), rapeseed oil (RO) (refined rapeseed oil, 4.99% of saturated fat, 74.11% of monosaturated fat and 20.90% of polyunsaturated fat; Mazola brand, UK retail market) and sunflower oil (SO) (refined sunflower oil, 6.52% of saturated fat, 31.95% of monosaturated fat and 61.53% of polyunsaturated fat; Brakes Bros Co., Ltd., Kent, UK). The fatty acid composition of oils was analysed following the methodology described in Section 2.4.1 and is available in Appendix A.

### 2.3. Emulsion Preparation 

The conventional emulsion preparation procedure was based on the method described by Arancibia et al. [39] and Taha et al. [40] with some modifications. Conventional emulsions (oil droplet particle size of more than 200 nm) [35,36] were prepared in three steps. Firstly, a magnetic stirrer (Model SS3H, ChemLab, Zedelgem, Belgium) was used to disperse Tween 20 (5% *w*/*w*) in water (85% *w*/*w*) at 3.33 s^−1^ for 30 min at ambient temperature for complete dispersion. Then, the oil was added (10% *w*/*w*) to the aqueous phase during continuous stirring. Secondly, the emulsions were homogenised with a high-speed homogeniser (Model L4RT, Silverson, Bucks, UK) at 166.67 s^−1^ for 10 min. Thirdly, the emulsions were further processed in an ultrasound processor (32 mm diameter titanium probe, Model P100/6-20, Sonic Systems Limited, Ilmister, UK) at 100 watt, 20 kHz frequency, 5 microns amplitude at ambient temperature for 15 min (total power density delivered to the samples was 99 watts/L; power density was 12 watts/cm^2^). Then, the emulsions were left to cool down at ambient temperature and kept at 25 °C and 40 °C for 1, 7, 14, 21 and 28 days before measurements. All conventional emulsions were prepared in triplicate.

Nanoemulsions were produced after selecting three of the oils (EVOO, RO and SO). For nanoemulsion preparation, the coarse emulsions were processed in a high-pressure homogeniser (8.30H, Rannie, APV, Albertslund, Denmark) at 50,000 kPa for 1 cycle to produce droplet sizes < 200 µm. Then, the nanoemulsions were left to cool down at ambient temperature before measurements for 24 h. All nanoemulsions were prepared in triplicate.

The nomenclature used for the oil samples was as follows: extra virgin olive oil (EVOO), cold-pressed rapeseed oil (CPRO), olive oil (OO), rapeseed oil (RO), sunflower oil (SO). Conventional emulsion samples were named by adding a ‘-E’ after the oil name; nanoemulsion samples were named by addition of an ‘-NE’ after the oil name. When using the term ‘emulsions’, the authors refer to both conventional and nanoemulsions.

### 2.4. Physical and Chemical Properties of Oils

#### 2.4.1. Fatty Acid Composition 

The fatty acid composition of oils was determined according to Association of Analytical Communities method 969.33 [41]. Fatty acid methyl esters (FAMEs) were prepared by adding oils (200 mg) to a 0.5 M methanolic sodium hydroxide solution (4 mL). Then, the solutions were attached to a condenser and refluxed for 10 min until fat globules disappeared. Boron trifluoride solution (5 mL) was added, and the mixture continued boiling for 2 min. Heptane (5 mL) was added through the condenser, and the mixture was boiled a further minute. After boiling, the mixture was allowed to be tepid by keeping it at room temperature for 2 min, and saturated sodium chloride solution was added. Subsequently, the heptane layer was transferred into the test tube, and anhydrous sodium sulphate was added in order to remove the water. For the heptane phase, the solution was diluted with heptane to a 10% concentration, and 1 µL was injected in a gas chromatographer (GC; Agilent 7890B gas chromatograph (Agilent Technologies Ltd., Didcot, UK) equipped with a flame ionisation detector (FID) and an HP-5 capillary column (30 m × 0.32 mm i.e., 0.25 µm film thickness) (Agilent Technologies Ltd., Didcot, UK) for analysis.

GC conditions were set up following the method described by Nhu-Trang et al. [42]. The column temperature was programmed from 70 °C (held for 3 min), then increased up to 166 °C at 3 °C/min rate and to 285 °C at 15 °C/min. Injector and detector temperatures were 250 °C and 300 °C, respectively. Split ratio injection was 1:50. Helium was used as carrier gas at a flow rate of 1.5 mL/min. The relative percentage compositions of fatty acids were computed by normalisation method from the GC peak areas and calculated as the mean value of three injections.

#### 2.4.2. Total Phenolic Content

The total phenolic content (TPC) of oils was determined by using Folin–Ciocalteau colourimetric method according to Lee et al. [43] with some modifications. Oil samples (0.5 g) were extracted with methanol (10 mL) using a vortex mixer for 1 min. Then, 5 mL of the Folin reagent (previously diluted 10-fold with water) was added to 2 mL of the methanolic extract. The solution was incubated at room temperature for 5 min, followed by adding 7% (*w*/*v*) sodium carbonate (1 mL) and incubating for 90 min at room temperature. The absorbance was read at 725 nm using a spectrophotometer (CECIL, CE 1021, 1000 SERIES, Cambridge, UK). A calibration curve was constructed using gallic acid as standard, and the results were expressed as gallic acid equivalents (mg GAE/kg oil). A stock solution of gallic acid (0.5 mg/mL) was dissolved in 1 mL of methanol before diluting with distilled water to prepare a calibration curve of concentrations of gallic acid standards between 0.001 to 0.250 mg/mL. Measurements were taken in triplicate for each oil sample.

#### 2.4.3. Free Fatty Acids

The free fatty acids were determined using the official cold volumetric titration method [44,45] with potassium hydroxide. The samples (2.5 g) were dissolved in 50 mL of neutralised mixture of diethyl ether and ethanol 95% (*v*/*v*). The mixture was titrated with potassium solution (0.1 mol/L) and using 0.3 mL of phenolphthalein solution (10 g/L solution in 95% ethanol (*v*/*v*)) per 100 mL of mixture as an indicator. Measurements were taken in triplicate. Results were expressed as percentage of oleic acid (C18:1) and calculated using the following equation (Equation (1)):(1)$$FFA\;\left(\%\right)\;=\frac{V\times c\times M}{10\times m}\;$$
where V = the volume of titrate potassium hydroxide solution used, in millilitres,c = the exact concentration in moles per litre of the titrated solution of potassium hydroxide used,M = the molar weight in grams per mole of the acid used to express the result (oleic acid = 282) andm = the weight in grams of the sample.

#### 2.4.4. Emulsion Colour

The colour of the emulsions was assessed using a Chroma Meter CR-400 (Konica Minolta, Inc., Tokyo, Japan). The results were express in accordance with the CIELAB system with reference to illuminate D65 and a visual angle of 10°. The parameters determined were *L** (*L** = 0 (black) and *L** = 100 (white)), *a** (−*a** = greenness and +*a** = redness), *b** (−*b** = blueness and +*b** = yellowness). The samples were poured into a granular-materials attachment CR-A50 (Konica Minolta, Inc., Tokyo, Japan) and measured in triplicate.

#### 2.4.5. Determination of Radical Scavenging Activity

The antioxidant property of oils was evaluated by the 1,1-diphenyl-2-picrylhydrazyl (DPPH) assay. The DPPH radical from the odd electron of nitrogen atom can be scavenged by receiving a hydrogen atom from antioxidant compounds [46]. When the DPPH stable free radical was reduced, a change of colour from violet to yellow was observed. The oil samples were analysed for their radical scavenging activity following the method described by Mraihi et al. [47] with some modifications. As a reagent, a 0.1 mmol solution of 1,1-diophenyl-2-picry-hydrazyl (DPPH) was prepared in 80% methanol, and 5 mL of the solution was mixed with 0.25 mL of oil. This sample was incubated in the dark for 30 min. Then, the absorbance was measured at 517 nm against a control sample (5 mL of 0.1 mmol DPPH in methanol with 0.25 mL of blank solution). Measurements were taken in triplicate. The radical scavenging activity was expressed as the inhibition percentage of free radical DPPH, calculated using the following equation (Equation (2)):(2)$$\mathrm{DPPH}\cdot \mathrm{scavenging}\;\mathrm{activity}\;\left(\%\right)\;=\;\left(1-\frac{As}{Ac}\right)\;\times \;100$$
where Ac is the absorbance of the control, and As is the absorbance of the sample.

#### 2.4.6. Determination of Thiobarbituric Acid Reactive Substances (TBARS)

TBARS were determined according to the method of Qiu et al. [48] and Sharif et al. [49] with some modifications. Briefly, 0.1 mL of the oil sample was added to 5 mL of thiobarbituric acid (TBA) solution, which was prepared by mixing 15 g of trichloroacetic acid (TCA), 0.375 g of TBA and 2.1 g hydrochloric acid (37% *w*/*w*). Samples were heated in a water bath at 95 °C for 10 min; then, the samples were allowed to cool down to room temperature for 10 min, followed by centrifugation (Heraeus Multifuge 3SR Plus Centrifuge, Thermo Fisher Scientific Ltd., Waltham, MA, USA) at 10,000× *g* for 15 min. The absorbance of the supernatant was measured at 532 nm using a UV spectrophotometer (CECIL CE 1021 1000 Series, Cecil Instruments Ltd., Cambridge, UK). The concentrations of TBARS values were determined by using a standard curve prepared using 1,1,3,3-tetraethoxypropane (TEP) standard curve (coefficient correlation (R^2^) = 0.9994). TEP standards between 0.01 to 0.20 µg/mL were prepared with trichloroacetic acid 7.5%. Three analytical repetitions of each measurement were performed for each emulsion batch.

#### 2.4.7. Viscosity

The viscosity of oils was measured with a rheometer (MCR 102, Anton Paar, St. Albans, UK) with a concentric cylinder (CC27, Anton Paar, St Albans, UK). The shear rate range used was from 0.1 s^−1^ to 200 s^−1^, and the temperature was maintained at 25 °C. Measurements were taken in triplicate. The viscosity values at 100 s^−1^ were taken for data analysis and comparison.

#### 2.4.8. Density

The apparent density (weight by volume) of oil samples was determined by the method followed by Gunstone [5]. Oils’ weight and volume were measured at room temperature, and apparent density was calculated using the following equation (Equation (3)):(3)$$\mathrm{Apparent}\;\mathrm{density}\;\left(g/\mathrm{mL}\right)\;=\;\frac{Mass\;of\;oil\;\left(g\right)}{Volume\;of\;oil\;\left(\mathrm{mL}\right)}$$

### 2.5. Physical and Chemical Properties of Conventional and Nanoemulsions

#### 2.5.1. Measurement of Emulsion Mean Droplet Diameter (MDD) and Polydispersity Index (PDI) 

Particle size and polydispersity index of emulsions were determined in a dynamic light scattering (DLS) instrument (Zetasizer Nano ZS, Malvern Instruments Ltd., Worcestershire, UK) by following a method described in previous studies [49,50]. Emulsions were diluted 100-fold with deionised water and agitated in order to avoid multiple light scattering effects. The dispersion was decanted into polystyrene cuvettes, and the measurement was recorded at a wavelength of 633 nm at 25 °C. Measurements were taken in triplicate. The polydispersity index was calculated with the following equation (Equation (4)) [51]:(4)$$\mathrm{PDI}={\left(\sigma /d\right)}^{2\;\;}$$
where σ is the standard deviation, and d is the mean particle diameter.

#### 2.5.2. Creaming Index (CI) and Thermal Stability (TS)

Creaming index (%) was evaluated based on the method reported by previous studies with some modifications [39]. An amount of 10 mL of each emulsion was poured into a glass tube and stored at 25 °C and 40 °C in order to accelerate destabilisation mechanisms during storage. The total height (mm) of emulsion and cream layer were measured with a digital caliper after 1, 7, 14, 21 and 28 days. The Cl (%) was calculated using the following equation (Equation (5)):(5)$$\mathrm{Creaming}\;\mathrm{index}\;\%=\;\left(\frac{Hc}{Ht}\right)\times 100\;$$
where Ht is the total height of the emulsion (mm), and Hc is the height of cream layer (mm).

Thermal stability was determined as described by Sahafi, Goli, Kadivar and Varshosaz [31]. Each emulsion (10 mL) was heated in a water bath at 80 °C for 30 min, followed by centrifugation at 1200× *g* for 10 min. The height (mm) of initial emulsion, cream layer and sedimentation phase was measured with a Digital Vernier Caliper. Emulsion thermal stability was calculated according to Equation (6):(6)$$\mathrm{Thermal}\;\mathrm{stability}\;\%=\;\left(\frac{HE\;-\;\left(HS\;+\;HC\right)}{HE}\right)\times 100$$
where HE is the height of initial emulsion (mm), HS is the height of sedimentation phase (mm), and HC is the height of cream layer (mm).

#### 2.5.3. Determination of Thiobarbituric Acid Reactive Substances (TBARS)

The measurement of TBARS was performed as explained in Section 2.4.6, with a different sample/reagent proportion [48]. An amount of 1 mL of emulsion was added to 5 mL thiobarbituric acid (TBA) solution and measured after 1, 7, 14, 21 and 28 days at stored temperature of 25 °C and 40 °C. Measurements were taken in triplicate.

### 2.6. Statistical Analysis

Statistical analysis of experimental data was performed using IBM SPSS 25 (IBM Corp, Armonk, NY, USA). One-way analysis of variance (ANOVA) and Tukey’s test at 95% confidence level (*p* < 0.05) were used to compare the mean values of viscosity, density, total phenolic components, free fatty acids, radical scavenging activity and TBARS of oil samples, and MDD, PDI, ζ-potential, creaming index and TBARs of emulsion samples. Moreover, to evaluate the effect of storage time in conventional emulsions, a two-way analysis of variance and Tukey’s test at 95% confidence level (*p* < 0.05) were conducted. The interaction of the two independent factors, oil type (EVOO, CPRO, OO, RO, SO) and storage time (1, 7, 14, 21 and 28 days), at two different storage temperatures (25 °C and 40 °C), was evaluated for the creaming index and TBARS values of conventional emulsion samples. Pearson correlation was calculated for the physical and chemical properties of vegetable oils, conventional, and nanoemulsions at 95% and 99% confidence level (*p* < 0.05 and 0.01, respectively). The correlation coefficient (r) was obtained: very weak correlation (0.01 ≤ r < ±0.10), weak (±0.10 ≤ r < ±0.50), moderate (±0.50 ≤ r < ±0.80), strong (±0.80 ≤ r < ±1.00) and perfect (r = ±1.00) [52].

## 3. Results and Discussion

### 3.1. Effect of Oil Type on the Stability of Conventional Emulsions

#### 3.1.1. Mean Droplet Diameter (MDD) and Polydispersity Index (PDI)

Droplet size influences the physicochemical stability of emulsions [7,53]. A smaller droplet size could lead to better stability against droplet coalescence and flocculation because of the reduction in Brownian motion and gravitation forces [53], which could lead to a slower rate of creaming compared to larger droplet size. Regarding polydispersity index (PDI), it describes the particle size distribution of droplets, which is closely related to the stability of the emulsion [8]. Low PDI values represent a narrower distribution of particle size, therefore, a small difference in droplet size. Emulsions with narrower PDI values showed to be more stable than emulsions with wider PDI [54] due to a lower Ostwald ripening [53,55]; Laplace pressure inside smaller droplets is higher than in larger droplets, resulting in smaller droplets merging with larger droplets [56]. The results of the MDD and PDI of conventional emulsions are presented in Table 1. The results showed that there were no significant differences in droplet size and PDI (*p* ≥ 0.05) among emulsions. The formation of emulsions could be mainly affected by ingredient composition (oil, surfactant, water) [29], ratio of components [57] and, significantly, by process parameters [29], such as energy intensity and sonication duration [58,59]. The PDI values of conventional emulsions exhibited a wide distribution of particle size (more than 0.4), indicating that the samples were of low stability. PDI values below 0.2 indicate a good stability of the colloid suspension [60]. For ultrasonication, a decrease in droplet size and PDI of emulsions is created by cavitation with high-energy sound waves [61], which depends on frequency, amplitude and sonication process [62].

In the present study, the only variable factor was oil type with constant oil and emulsifier concentration, which did not have a significant effect on droplet size and PDI. This observation is similar to the findings of previous studies [31,49], which reported that PDI of nano/conventional emulsions formulated with constant emulsifier concentration was not influenced by oil type. The required hydrophilic lipophilic balance (HLB) of oils plays an important role in mean droplet diameter and stability of emulsions [63,64]. When HLB values of surfactants or combinations of surfactants are close to the required HLB values of the oil phase, more stable emulsions will be formulated [63,64,65]. The required HLB value of vegetable oils, such as olive oil, rapeseed oil, palm oil, safflower oil and sesame oil, is 7 [66,67]. The similarity in the HLB value required for all the oils studied could have led to similar emulsion formation ease, droplet size and PDI of emulsions. Therefore, these results indicated that the type of oil with high content of unsaturated fatty acids has a negligible effect on MDD and PDI.

ζ-potential is the electrokinetic potential difference between the dispersion medium and the stationary layer of fluid attached to the dispersed particle [68]. ζ-potential is an important parameter because it is used to predict and control the stability of emulsions, indicating the degree of repulsion between adjacent, similarly charged particles in dispersion [68]. As can be observed in Table 1, all samples had similar (*p* > 0.05) negative ζ-potential values. High ζ-potential values (>+30 or −30) are related to better stability of the dispersed particles [69]. Although all emulsions were stabilised with Tween 20 (polyoxyethylene sorbitan monolaurate), a nonionic surfactant, the negative ζ-potential of emulsions could be due to the presence of ionic impurities, such as free fatty acids, or adsorption of hydroxyl (OH^−1^) to the surface of droplets [28,31,70].

The colour results of the conventional emulsions are presented in Table 1 and Figure 1. The lightness (*L**) of the emulsion can be influenced by the oil type used (Appendix A), but also by the droplet size, the concentration of droplets and chromophore compounds [71]. In the present study, the lightness of EVO-E was significantly (*p* < 0.05) lower than in other oils (Table 1). These results could be related to a higher concentration of chlorophyll in EVOO [72] than in other samples studied. Lightness decreases with increasing concentration of chromophore compounds due to a higher light absorption of chromophoric molecules, which allow less light being transmitted from emulsions [73]. The highest levels of greenness and yellowness (*p*
< 0.05) were found in EVO-E and CPR-E, respectively, and they were related to the presence of pigments in the oil, such as chlorophyll and carotenoids [72,74].

#### 3.1.2. Creaming Index (CI) 

Creaming index (CI) is a method of measuring gravitational separation in emulsions [8]; low CI (%) values are related to more stable emulsions. In this study, the emulsions were stored at 25 °C and 40 °C for 28 days to evaluate their physical stability over time. No significant interaction (*p* > 0.05) between oil type and storage time was observed at both storage temperatures (25 °C and 40 °C). However, both factors (oil type and storage time) showed a significant effect on CI values (Figure 2a,b). EVO-E, CPR-E and SO-E showed a significantly (*p* < 0.05) lower CI value than RO-E, which showed the highest CI value (Figure 2a). The CI values of emulsions can be influenced by various factors, including droplet size and density of the dispersed (oil) phase. It can be supported by Stokes’ law that the rate of gravitational separation can be decreased by a decrease in droplet size, a decrease in density difference between the dispersed phase and the continuous phase and by an increase in viscosity of the continuous phase [8]. As a consequence, the lower CI value of SO-E compared to RO-E at both 25 °C and 40 °C could be due to a significantly lower density difference between the oil phase and the water phase (Appendix A), and the largest difference in the droplet size between SO-E and RO-E compared to other oils, where the size of SO-E was the smallest, whereas the biggest size was found in RO-E (Table 1). A decrease in droplet size results in a decrease in the attractive forces between the droplets [53,75]; smaller droplets have better stability against droplet coalescence and flocculation because of the reduction in Brownian motion and gravitation forces [53], resulting in a slower rate of creaming layer. On the other hand, an increase in storage time showed a significant increase (*p* < 0.05) in CI values at both temperatures over the period of time investigated (Figure 2b). Oil droplet aggregation, including flocculation and coalescence, takes place over time and leads to creaming layer formation in emulsions [76]. This was in agreement with previous studies that reported a higher level of creaming index of emulsion when the storage period increased [77,78]. Although there was a similar trend of Cl values at 25 °C and 40 °C, the emulsions stored at 40 °C showed faster destabilisation than those stored at 25 °C, indicating that the aggregation rate of oil droplet particles was greater at higher temperature, leading to a higher value of creaming index. This result could be due to the density of the dispersed phase (oil), which decreases with increasing temperature [79], which could lead to an increase in density difference between dispersed phase and continuous phase, resulting in a higher rate of CI value.

#### 3.1.3. Determination of Thiobarbituric Acid Reactive Substances (TBARS)

TBARS was used to measure the secondary products of lipid oxidation in emulsions in order to investigate the effect of oil types on lipid oxidative stability over time. In the present study, the emulsions were kept at 25 °C and 40 °C after formation, and TBARS lipid oxidation value was measured at 1, 7, 14, 21 and 28 days storage time. At 25 °C storage temperature, there was a significant interaction (*p* < 0.05) between oil type and storage time when evaluating TBARS (Figure 3a). EVO-E, OO-E and SO-E presented significantly lower (*p* < 0.05) TBARS values than CPR-E and RO-E during the storage time. In contrast, CPR-E and RO-E showed a significant increase in TBARS values at 1, 7, 14 and 21 days. The highest TBARS values were found in CPR-E at 28 days storage. At 40 °C storage temperature, there was a significant interaction between oil type and storage time for the TBARS values (Figure 3b). At 40 °C, TBARS values were higher and increased more rapidly than at 25 °C. Storage at 40 °C accelerated lipid oxidation, leading to higher TBARS values in all emulsions during storage. The lipid oxidative stability of the emulsions presented a similar trend as at 25 °C; EVO-E, OO-E and SO-E showed significantly lower TBARS values than CPR-E and RO-E. These three oils showed a greater stability during storage with no significant differences in TBARS values after 21 days of storage. On the other hand, as storage time increased, TBARS values significantly increased (*p* < 0.05) for RO-E and CPR-E. The highest TBARS value was shown in RO-E at 28 days. The significantly lower TBARS values in SO-E, OO-E and EVO-E than in CPR-E and RO-E could be related to differences in the fatty acid composition of the oils (Appendix A). Emulsions formulated with oils with higher fraction of unsaturated fatty acids (CPRO, RO) lead to higher oxidation rates because unsaturated fatty acids are more susceptible to oxidation than saturated fatty acids. Unsaturated fatty acids have a double bond in the carbon chain; the hydrogen atom attached to the carbon between the double bond is easily removed and provides alkyl radicals [80]. Similar results were also reported by Kiokias, et al. [81] in a study where they observed that olive kernel oil emulsion was oxidised less than corn oil, cottonseed oil and sunflower oil emulsions because olive kernel oil contained less polyunsaturated fatty acids than the other oil.

The correlations between emulsion stability and the type of oil were further assessed. Pearson’s correlation coefficients (r) between the physical and chemical properties of oils (USFA, TPC, antioxidant capacity and FAA) and the stability of emulsions (creaming index and TBARs) are presented in Table 2. USFA exhibited significant positive correlations with creaming index (*p* < 0.05) and TBARs (*p* < 0.01) of emulsions. The moderate positive correlation for creaming index and TBARs suggested that vegetable oils with higher content of unsaturated fatty acids exhibited higher values of creaming index and TBARs, indicating that fatty acid composition of oils affects the stability of emulsions. The higher proportion of unsaturated fatty acids in oils leads to higher rates of lipid oxidation [80]. Refined oils, such as SO and RO, showed lower values of FFA than unrefined oils, such as EVOO and CPRO. In terms of SO, the crude sunflower oil is processed through the conventional alkali refining process, which consists of degumming, alkali neutralisation, dewaxing, bleaching and deodorisation stages [82]. Therefore, the lower TBARS value of SO-E among the emulsions could be due to a lower content of free fatty acids in the oil (Appendix A), which could be due to the refining process of oil, especially the neutralisation and deodorisation stages in which free fatty acids are removed [22]. Free fatty acids can work as prooxidants to accelerate the decomposition of lipid hydroperoxides, enabling free radicals to form a secondary lipid oxidation product [83]. Another factor influencing the emulsion oxidative stability is the presence of antioxidant components in oils, such as in EVOO and OO (Appendix A) [19], which can lead to lower TBARS values in EVO-E and OO-E (Figure 3). The phenolics and tocopherols in oils can scavenge the free radicals and contribute to the antioxidant activity. Previous studies have also shown a positive correlation between the phenolic and tocopherol content and radical scavenging activity [13,17]. As a consequence, EVO-E, OO-E and SO-E were more stable than CPR-E and RO-E. However, it can be seen from Table 2 that no significant correlation was found between TBARs and radical scavenging activity of oils. This was in agreement with previous results, where it was reported that no significant correlation was found between lipid oxidative stability and natural antioxidant content, such as tocopherol [81]. These results suggest that the properties of oils, including fatty acid profile, free fatty acids content and antioxidant capacity, are important factors influencing the emulsions’ TBARs values and should be considered when formulating emulsions. Therefore, in order to evaluate the effect of oils on the properties of conventional and nanoemulsions oils that have different fatty acid profiles, the content of free fatty acids, total phenolic compounds and a high oxidative stability were selected (EVOO, RO and SO).

### 3.2. Comparison between Conventional and Nanoemulsions

Conventional and nanoemulsions were formulated with EVOO, RO and SO. Correlations between MDD, PDI, thermal stability and TBARs were evaluated for those samples. As shown in Table 3, MDD and PDI of nanoemulsions presented significantly lower values (*p* < 0.05) than conventional emulsions. When using high-pressure homogenisation to produce nanoemulsions, there is an increase in shear force and cavitation, which results in a reduction of particle size and polydispersity values [75,84]. Smaller droplet size could lead to greater physical stability [53] and higher lipid oxidation of nanoemulsions due to an increased droplet surface area [7,85]. It was observed that the colour of nanoemulsions was lighter than that of conventional emulsions. These results could be explained, as the scattering efficiency of droplets decreases when the droplet size increases [86]. Correlation coefficients between MDD, PDI and stability of conventional and nanoemulsions are presented in Table 4. MDD and PDI showed a significant strong negative correlation (*p* < 0.01) with CI, whereas there were insignificant correlations with TBARs. The negative correlation suggested that emulsions with smaller droplet size and narrower size distribution values exhibited higher thermal stability. This correlation could be explained by Stokes’ law; the rate of gravitational separation can be decreased by a decrease in droplet size [8]. A decrease in particle droplet size enabled a better stability against droplet coalescence and flocculation due to the reduction in Brownian motion and gravitation forces [53], and a decrease in the attractive forces between the droplets [53,75]. Moreover, the narrower PDI values of nanoemulsions lead to more stable colloids (Table 2) due to a lower Ostwald ripening; there is a diffusive migration of smaller droplets with higher Laplace pressure to larger droplets [53,55,56]. In summary, MDD and PDI were an important influence on the thermal stability of conventional and nanoemulsions.

## 4. Conclusions

This study investigated the influence of vegetable oils from different natural origins and production processes on the physicochemical properties and stability of emulsions. The selected vegetable oils used in this study showed differences in the level of unsaturation total phenolic content, free fatty acids and antioxidant activity, which could be due to the difference in refining and extraction processes used in oil production. The results showed that the physicochemical stability of the emulsion was affected by fatty acid composition, the presence of antioxidants, free fatty acids and droplet size in oils. SO-E, OO-E and EVO-E showed significantly higher lipid oxidative stability compared to RO-E and CPR-E after 28 days storage time due to a higher fraction of unsaturated fatty acids in CPRO and RO. The content of free fatty acids and total phenolic compound in the oil could also be an important factor influencing the emulsions’ TBARs values. Nanoemulsions showed greater physical stability than conventional emulsions, mainly when formulated with EVOO due to its higher radical scavenging activity and oxidation stability. However, there were insignificant correlations between oxidative stability of emulsions, and oil droplet size and antioxidation capacity of oil. Overall, this study provides valuable insights into the characteristics of refined and unrefined oils with high content of unsaturated fatty acids and their effects on the physicochemical properties of both conventional emulsions and nanoemulsions. This information is important for the selection of suitable oils to develop emulsion with enhanced nutritional fatty acid profile and desirable characteristics for application in food products [87].

## Figures and Tables

**Figure 1 foods-11-00681-f001:**
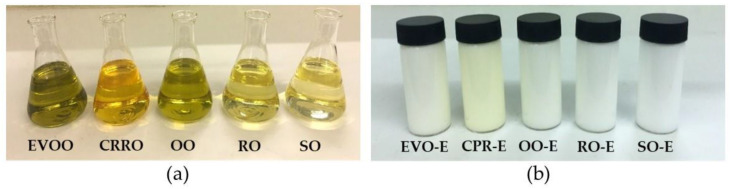
Visual appearance of (**a**) vegetable oils, (**b**) conventional emulsions (-E) prepared with the different oils: extra virgin olive oil (EVO-E), cold-pressed rapeseed oil (CPR-E), olive oil (OO-E), rapeseed oil (RO-E), sunflower oil (SO-E).

**Figure 2 foods-11-00681-f002:**
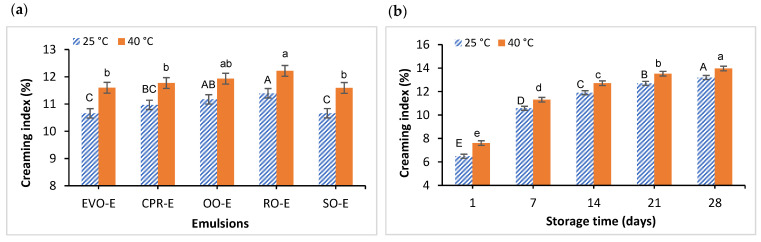
Creaming index (%) means values and 95% confidence intervals of the conventional emulsions stored at 25 °C and 40 °C. (**a**) Effect of oil type; (**b**) effect of storage time. Different capital letters indicate significant differences between mean at 25 °C, and different lower-case letters indicate significant differences between means at 40 °C. Conventional emulsions (-E) prepared with the different oils: extra virgin olive oil (EVO-E), cold-pressed rapeseed oil (CPR-E), olive oil (OO-E), rapeseed oil (RO-E), sunflower oil (SO-E).

**Figure 3 foods-11-00681-f003:**
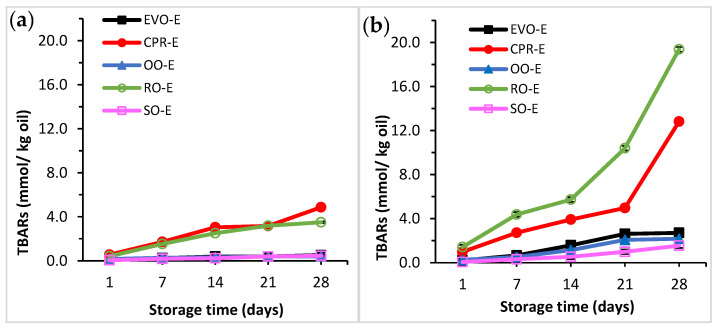
TBARS of conventional emulsions during storage time at (**a**) 25 °C and (**b**) 40 °C. Conventional emulsions (-E) prepared with the different oils: extra virgin olive oil (EVO-E), cold-pressed rapeseed oil (CPR-E), olive oil (OO-E), rapeseed oil (RO-E), sunflower oil (SO-E).

**Table 1 foods-11-00681-t001:** Mean droplet diameter (MDD), polydispersity index (PDI), ζ-potential and colour of conventional emulsions (-E) prepared with the different oils: extra virgin olive oil (EVO-E), cold-pressed rapeseed oil (CPR-E), olive oil (OO-E), rapeseed oil (RO-E), sunflower oil (SO-E).

Conventional Emulsions	MDD (nm)	PDI	ζ-Potential (mV)	Colour		
				*L**	*a**	*b**
EVO-E	247.57 ^a^ (9.32)	0.448 ^a^ (0.038)	−32.42 ^a^ (0.48)	77.30 ^c^ (2.64)	−4.78 ^e^ (0.08)	8.69 ^b^ (0.57)
CPR-E	249.60 ^a^ (4.60)	0.472 ^a^ (0.050)	−31.69 ^a^ (2.35)	81.63 ^b^ (1.56)	−3.32 ^d^ (0.17)	14.85 ^a^ (0.27)
OO-E	238.23 ^a^ (8.75)	0.449 ^a^ (0.040)	−29.79 ^a^ (1.55)	83.61 ^ab^ (1.96)	−2.29 ^c^ (0.07)	3.83 ^c^ (0.20)
RO-E	251.94 ^a^ (7.69)	0.495 ^a^ (0.049)	−31.30 ^a^ (0.67)	85.78 ^ab^ (0.34)	−1.51 ^b^ (0.01)	0.76 ^d^ (0.12)
SO-E	239.41 ^a^ (8.20)	0.474 ^a^ (0.040)	−31.81 ^a^ (1.12)	85.04 ^a^ (1.47)	−1.04 ^a^ (0.04)	−0.58 ^e^ (0.04)

Indicated values are reported as means (standard deviation). Values with the different superscript letters are significantly different (*p* < 0.05) between samples in the same column.

**Table 2 foods-11-00681-t002:** Pearson’s correlation coefficients between the unsaturated fatty acids content (USFA), total phenolic content (TPC), radical scavenging activity, free fatty acids (FFA) of oils and the creaming index (CI) and thiobarbituric acid reactive substances (TBARS) conventional emulsions at 25 °C.

Variable	USFA (%)	TPC (mg GAE/kg Oil)	Radical Scavenging Activity (%)	FFA (% Oleic Acid)	CI (%)	TBARs (mmol/kg Oil)
USFA	1	-	-	-	-	-
TPC	−0.749 **	1	-	-	-	-
Antioxidant	−0.277	0.745 **	1	-	-	-
FAA	−0.632 *	0.759 **	0.287	1	-	-
Creaming index	0.592 *	−0.208	0.232	−0.255	1	-
TBARs	0.741 **	−0.356	−0.063	−0.103	0.789 **	1

*, ** Indicated that the correlation is significant at *p* < 0.05 and 0.01, respectively.

**Table 3 foods-11-00681-t003:** Mean droplet diameter (MDD), polydispersity index (PDI), creaming index (CI) and thiobarbituric acid reactive substances (TBARs) of conventional and nanoemulsions at 25 °C.

Emulsion Type	Vegetable Oils	MDD (nm)	PDI	CI (%)	TBARs (mmol/kg Oil)
Conventional emulsions	EVOO	467.60 ^a^ (66.23)	0.928 ^ab^ (0.122)	89.49 ^b^ (0.25)	0.31 ^c^ (0.02)
	RO	452.02 ^ab^ (69.76)	0.890 ^b^ (0.134)	89.00 ^b^ (0.29)	2.33 ^b^ (0.07)
	SO	401.09 ^b^ (14.66)	1.000 ^a^ (0.000)	89.42 ^c^ (0.36)	0.20 ^d^ (0.03)
Nanoemulsions	EVOO	192.19 ^c^ (3.28)	0.261 ^c^ (0.011)	100.00 ^a^ (0.00)	0.39 ^c^ (0.07)
	RO	192.86 ^c^ (4.08)	0.249 ^c^ (0.017)	100.00 ^a^ (0.00)	2.85 ^a^ (0.10)
	SO	195.02 ^c^ (3.37)	0.258 ^c^ (0.017)	100.00 ^a^ (0.00)	0.20 ^d^ (0.06)

Indicated values are reported as means (standard deviation). Values with the different superscript letters are significantly different (*p* < 0.05) between samples in the same column.

**Table 4 foods-11-00681-t004:** Pearson’s correlation coefficients between mean droplet diameter (MDD), polydispersity index (PDI), creaming index (CI) and thiobarbituric acid reactive substances (TBARs) of conventional and nanoemulsions.

Variable	MDD (nm)	PDI	CI (%)	TBARs (mmol/kg Oil)
MDD	1	-	-	-
PDI	0.899 **	1	-	-
Thermal stability	−0.938 **	−0.978 **	1	-
TBARs	−0.054	−0.138	0.097	1

** Indicated that the correlation is significant at *p* < 0.01.

## Data Availability

The data presented in this paper are openly available in the University of Reading Research Data Archive at https://doi.org/10.17864/1947.000361 (19 February 2022).

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
