# Peer review of "Physical and Chemical Characterisation of Conventional and Nano/Emulsions: Influence of Vegetable Oils from Different Origins"

_foods, 2022, doi:10.3390/foods11050681_

Round 1
Reviewer 1 Report
Dear Authors,
The Manuscript is well presented however, more details in the section materials and methods need to be added. In specific:
How can you fix and control the temperature during ultrasound treatment?
Could you please calculate the real ultrasound power for each sample?
Regarding the Figures: Please, improve the quality of Figures 1 and 2.
Table 1. Are you sure that particle size with an average diameter higher than 200 nm is still considered as nanoemulsions? Could you please provide a definition of this?
Thanks in advance.
Best regards,
Author Response
Please see the attachement.

Reviewer 2 Report
Overall, this study provides valuable insights into the characteristics of
refined and unrefined oils with high content of unsaturated fatty acids and their effects on
the physicochemical properties of both conventional emulsions and nanoemulsions.
I recommend the following issues need to be properly addressed:
This statement on page 7, lines 302-303: "Therefore, this result indicated that the type of oils with high content of unsaturated fatty acids
have a negligible effect on MDD and PDI." should also be discussed in terms of the RHLB (Required HLB value) for the oils studied. Please check
the literature and find the RHLB values for the tested oils and include these into a paragraph discussing this.
This will give your findings a more riguruous fundamental base.
How was the polydispersity index (PDI) determined by the Zetasizer. Please include the formula so we know how this is calculated.
Also, it is known that the appearance of the conventional emulsions and the nanoemulsions are different. The former appear bluish while the latter appear white.
It would be useful to see a photograph of the nanoemulsions and the emulsions created.
I wonder if the MDD of roughly 200 nm is enough to call these nanoemulsions. To me, it seem that a MDD of 200 nm is still in the range of conventional emulsion.
Page 11, line 450: "enhance" should be "enhanced".
Author Response
Please, see the attachement.

Reviewer 3 Report
Dear authors
In this section you will find the comments related to the review of your research, I hope they will be useful for the improvement of this.
The abstact must contain the most relevant results of the investigation, it is necessary to attach the most convincing values ​​that allow understanding the resolution of the problem, not only the statistically significant differences.
The introduction is too general, it should address the novelty of the article, the text presented in this section does not demonstrate the impact of the research carried out. A clear example is that the formation of disperse systems (nanoemulsions), which is the central theme of the article, is not conclusively mentioned. The research hypothesis is not found. I suggest rewriting this section.
Materials and methods
Reagents and standards come before all other sections, you also need to add the purity of the materials, chemical formulas and if possible the lot.
The units must be in the international system, review the entire document and perform the respective conversions. Example rpm to s-1 and bar to kPa.
The methodology for the preparation of what is called in this document as conventional emulsions is by ultra-high agitation, which mentions 10,000 rpm. What criteria did the authors take to define it as conventional? Since ultra-high agitation is used to the formation of nanoemulsions, which is evidenced in Table 1.
The correct nomenclature for molarity is mol, not M.
Line 224, take care of the superscripts in s-1
For the calculation of density, was it considered that the support material must be at a constant weight to determine the weight?
In the section it refers to a significance level of 95%, but in the abstract and in the text, it also mentions a probability value of 0.01, clarify.
Section 3.1.1. Zeta potential is mentioned in Table 1, why is it not discussed? The PDI is also considerably high, explaining why this value was obtained, which was similar in all the samples.
Figure 1 changes to trend lines to visualize the trend obtained at different temperatures, the statistical analysis can be reflected in the text. The analysis must be carried out based on the particle size, why was the measurement not carried out at the time of storage of the particle size for its relationship with the rate of cremation?
For the measurements carried out, the temperature is taken into account in a prevailing way, I suggest that it be included in the title of the article.
In general, reinforce the analysis of the results by correlating them, there is a lack of strength in the depth of the analysis, thiobarbituric acid can be analyzed in greater depth with the antioxidant capacity tests, developing an analysis of the function that this parameter can provide in the preparation of emulsions.
The conclusions must respond to the general objective of the article, as well as to the hypotheses raised.
Author Response
Please see attachement

Reviewer 4 Report
Dear Authors,
your paper has been sent for my consideration. An important aspect is addressed in the manuscript. The entire experiment was properly planned, and the analytical methods used were properly described. They are generally described in great detail, even too much detail. Appropriate statistical analysis of the data allowed to draw the right conclusions.
MS is prepared well. However, I have a few comments that I have included in the PDF file.

Author Response
Please see attachement

Round 2
Reviewer 1 Report
No additional comments in the revised version.
Best,
Reviewer 3 Report
Dear authors
I appreciate the changes made to the document.
Regrads